# Missing the Present: Nostalgia and the Archival Impulse in Gentrification Photography

## Zeena Price

Tilburg School of Humanities, Department of Culture Studies, Tilburg University,
5037 AB Tilburg, The Netherlands; z.j.price@tilburguniversity.edu

**Abstract:** If gentrification is a violent form of "un-homing" (Elliot-Cooper et al., p. 494), then it is no surprise to witness an intensification of photographic practice in gentrifying areas; photography is, after all, fundamentally a place-making practice. Taking "home" to include the wider neighborhood and urban environment (Blunt and Sheringham 2019), this paper argues that the concept of anticipatory nostalgia is a useful way of understanding the recent wave of black and white photography in gentrifying areas. As well as signifying a sense of loss, anticipatory nostalgia, defined as missing the present before it has gone (Batcho and Shikh 2016), can also be seen as an aesthetic strategy of documenting places before they are lost to gentrification. Using the works of Colby Deal (*Beautiful, Still*), Jules Renault (*Suspended in Time*), and Lorenzo Grifantini (*W10*) as case studies, this paper argues that this type of photography, which explicitly utilizes an archival aesthetic, invites spectators to interrogate the intimate ties between home, memory, and identity. While melancholic, these images serve as a call to action and a form of speculation about the future—rejecting the shiny, computer-generated aesthetics of gentrification for a humanized, often gritty, and authentic version of home.

**Keywords:** gentrification; displacement; nostalgia; monochrome; archive

## 1. Introduction

If gentrification is a violent form of "un-homing" (Elliot-Cooper et al. 2019, p. 494), then it is no surprise to see an intensification of photographic practice in gentrifying areas; photography is, after all, fundamentally a place-making practice. In a context where home is made strange by aesthetic, social, and economic changes, photography can offer a sense of agency, visibility, and belonging and, potentially, create counternarratives of place and identity to those imposed by property developers and other vested interests.

Taking "home" to include the wider neighborhood and urban environment (Blunt and Sheringham 2019), this paper argues that the concept of *anticipatory nostalgia* is a useful way of understanding the recent wave of black and white photography in gentrifying areas. As well as signifying a sense of loss, anticipatory nostalgia, defined as missing the present before it has gone (Batcho and Shikh 2016), can also be seen as an aesthetic strategy of documenting places before they are lost to gentrification. This type of photography, as Grainge (1999) has argued, conveys a "nostalgia for the present", valorizing marginalized identities and places as objects worthy of cultural commemoration. Furthermore, in portraying experiences of gentrification through the lens of nostalgia, these images illuminate what Bondi (2006) calls the "emotional geographies" of gentrification, expanding understandings of displacement to include an important affective component. Anticipatory nostalgia, which is characterized by both a sense of loss *and* a sense of hope, adds nuance to concepts of displacement as "root shock", trauma, and grief.

In order to appreciate the affective power of the monochrome trend and its significance in the context of gentrifying places, I will provide some context on the issue of gentrification and how the concept of home—and un-homing—is central to the process. I then discuss

the concept of anticipatory nostalgia and its temporal ambiguity. What is it, and how does it function as both a "mood" or feeling and a "mode" or aesthetic convention? How might anticipatory nostalgia be understood as both melancholic and rooted in loss, while simultaneously serving as a call to action or as a speculation on urban futures? Finally, how radically does such an aesthetic depart from existing portrayals of gentrifying spaces? How, in short, do these images challenge us to reconceive of home in a context of gentrification? As well as engaging in a theoretical discussion, I also provide several images to illustrate these arguments. While these individual instances are both important and interesting, it is really the totality of black and white images or the popularity of the genre as a whole that I wish to convey. Note that the term "genre" itself is extremely controversial within literary and cultural theory, and what forms a genre to one scholar may not even qualify as a sub-genre to another. While common understandings of genre define it as a group of texts that share "family resemblances" or particular formal conventions, including themes, setting, or style (Chandler 1997), I prefer Swales' emphasis on *purpose* over substance or form. In this view, "the principal [criterion] that turns a collection of communicative events into a genre is some shared set of communicative purposes" (Swales 1981, p. 46 cited in Chandler 1997, p. 4). The images I discuss can all be characterized as social documentary photography and fall into the specific sub-category of monochrome. All take place in gentrifying communities for the self-proclaimed purpose of memorializing those places before they are gone, and all but one are the work of residents who live in those places. I have therefore designated "monochrome gentrification photography" as a genre in itself both to distinguish it from the vast swathe of social documentary photography that exists and to underline its popularity and significance. I turn first, however, to the research on gentrification and the lacuna within it: home.

## 2. Gentrification, Home, and Un-Homing (or Why Displacement Matters)

Gentrification studies has a problem. It seems to have suffered from a spectacular blind spot with regards to the issue of displacement, which can be conceived of as both a physical and symbolic loss of home (Atkinson 2015). While scholars can loosely agree on a definition of gentrification as urban population *change*, the degree to which *displacement* exists or is a relevant factor in gentrifying places is strongly disputed (Slater 2010). Some scholars, for example, argue that displacement is more accurately described as the replacement or upward mobility of working-class populations (Hamnett 2003), or that the scale on which true displacement occurs is nonexistent or marginal to the point of irrelevance (Freeman 2005). Other scholars have tried to rebrand the very term "gentrification" itself, arguing that neighborhood improvements bring benefits to poorer residents as well (Freeman and Braconi 2004 in Davidson 2009).

Others, however, have pushed back on such characterizations, arguing that not only does displacement deserve to be put in the front and center of gentrification research, but that it is a far more complex phenomenon than was previously thought to be the case (Davidson 2009; Elliot-Cooper et al. 2019). First of all, displacement is a deeply affective experience, akin to a form of trauma (Fullilove 2004) or grief (Fried 1966; Pain 2019). Secondly, previous discussions of displacement have been characterized by a severely reductive understanding of displacement as mere *physical* migration (Davidson 2009; Elliot-Cooper et al. 2019), ignoring the extent to which individuals can feel displaced *while remaining in the same location* (e.g., through the transformation of communal spaces or social changes that may leave them feeling disoriented; see Twigge-Molecey 2014). Displacement means more than just physical absence. It is not an end point—it emphasizes, rather, the slow and incremental changes that build up over time, leading to a lack of recognition and belonging to familiar surroundings. It is, as Lees argues, a violent "form of un-homing", and "a material and symbolic rupture" that "severs the connection between people and place" (Elliot-Cooper et al. 2019, p. 5).

This is the context, then, in which my interest in photographic practice emerges. In spaces of transience, of instability, of the loss of the familiar—all these give rise to an

impulse to document and preserve. Home is made up of the connections between people and places across time; gentrification erodes these ties, or at least puts them under strain. Photography in this environment becomes a way of holding on and forging connections, of making memories before they are transformed beyond recognition. It becomes a way of rooting people and of staking a claim to place. In the course of my research, I have noticed a remarkable tendency by photographers in different locations to turn to black and white modes of representation. Why? What can monochrome communicate that color cannot? What does it tell us about the temporalities of gentrification, and about the way those temporalities are linked to issues of memory, identity, and place?

In order to address these questions, I structure my discussion in the following way. First, I outline Paul Grainge's argument that the aesthetic choice of black and white works as a form of "visual remembrance" and is strongly linked to constructions of cultural memory (Grainge 1999). I then expand his arguments to discuss this form of memory-making in the *present* by using the concept of anticipatory nostalgia. This psychological construct, which here functions as an aesthetic mode, encompasses multiple temporalities and affective dimensions. On the one hand, it is melancholic, past-facing, and concerned with loss. This lends it an ambiguous temporal quality, with important implications for the concept of belonging. Yet it is precisely this quality of pastness—what Grainge calls an "archival aura"—that allows it to function as a site of reflection on cultural heritage and what is worth preserving in the present and future, ultimately lending these images a speculative and even hopeful quality. These images, in foregrounding nostalgia, thus highlight the affective dimensions of displacement and at the same time reject what Lindner and Sandoval (2021) refer to as consumption-led, exclusive, and exclusionary spaces of gentrification.

### 3. Archiving the Present: Black and White as/and the Aesthetics of Cultural Memory

Before discussing the importance of black and white to the formation of cultural memory, it is important to note that *all* photography, including in color format, is inherently linked to the past. Photographs, to repeat the frequently cited phrase by Roland Barthes, are artefacts that conjure once-present, now-absent, objects: the "what-has-been". Marita Sturken renders this argument eloquently when she writes that

> In its freezing of time in an instant, and its capacity to carry the image of the dead forward in time, the photograph renders a mortality to its subject. A photograph represents the what-has-been, awarding to its subject the quality of being of the past—once a photograph is taken, its moment is situated in the past . . . giv[ing] the photograph an aura of death. (Sturken 1999, pp. 190–1)

The perception of photography as conferring a "death" or finality on its subject is underscored by metaphors of "embalming" the moment of capture and rescuing it from the "corruption" of time (Bull 2010). Indeed, the influence of Barthes' mournful theory of photography as the "what has been", in which photographs function as objects of loss, cannot be overstated.

While this is (often accepted as) true of all photography, I would argue, following Grainge (1999), that it is especially true of black and white photography, which holds special connotations of pastness. First, black and white is strongly associated with its role as a historical technological development in photographic practice. Put simply, it is a technology of the past, recalling a time before color photography was invented (Boyd and Gorman-Murray 2023). For this reason, its use evokes a technological nostalgia in the viewer. However, second and more importantly, black and white occupies a special role in the *cultural* construction of the past by lending images "a status of authenticity" (Grainge 1999, p. 383). This "authenticity" relies on an archival aesthetic: it "helps construct narratives that gives issues and events the distance and authority of time; it has the potential for legitimation, *giving archival aura* to people and politics . . . " (385, italics mine).

### 4. Anticipatory Nostalgia—Archiving the Present before It Has Gone

I argue that this archival aura is key to understanding a recent wave of gentrification photography, in which places under threat are documented and preserved for cultural memory. As Grainge himself acknowledges, black and white has a special significance when used in the *present* as that which visually marks "what should be considered historic business" (1999, p. 392). As is the case when applied to genuine historical representations, it is a way of signifying a kind of historical legitimacy and cultural significance to subjects in the present. The monochrome image jumps from the realm of news or journalism to that of historical document; images are given meaning by their association with the archive. As Grainge puts it, "If past events gain meaning by their existence *in* history, one could also say that present events are given meaning by their identification *as* history" (388). This sense of significance is bolstered by an air of surrealism, even mysticism, that monochrome enjoys. We do not see in black and white; thus, every creative choice to use it immediately makes us step back and evaluate the object in question as something of unique and special status—something outside of ordinary experience (Freeman 2017).

Furthermore, this aesthetic practice, which functions as a form of visual remembrance, has a strongly future-oriented or *anticipatory* tendency, for it "help[s] construct the present as a future memory *being lived*, as an authentic past *in creation*" (388). Grainge references the future again later in his article, when he argues that "Essentially, the present is understood in terms of it being a future past ... we look forward to the future in order to look back on the present being lived, complete with shape and a sense of its archival place in historical narrative" (389).

Although he does not describe his ideas in these terms, I suggest that this nostalgia for the present parallels what Batcho and Shikh call "anticipatory nostalgia", a form of "mental time travel" in which there is a conflict between a hypothetical, imagined future and a "someday past" that is still present (Batcho and Shikh 2016, p. 75). This is fundamentally different from common understandings of nostalgia as relating to an object of loss in the *past*. The term nostalgia, literally defined as yearning for a lost home, originates from a combination of the Greek *nostos* (return home) and *algos* (pain or sorrow), resulting in feelings of loss but also the bittersweet pleasure of remembering (Boym 2001). Nostalgia has traditionally been given rather a bad press, coming to symbolize a kind of reactionary sentimentalism and paralysis or inertia, "a general sense of loss and regret, a kind of mourning for the impossibility of return" to a lost time or a lost place (May 2017, p. 404). Importantly, it seems to imply "a sense of unhappiness with the present" (ibid.) in which the past is idealized; it is "defeatist" and demonstrates a "failure to cope with change" (ibid.). Indeed, this is a criticism leveled against the heritage industry more broadly—that in attempting to freeze or preserve the present into a relic, future change becomes harder to enact. Put differently, if the present is communicated as already past, what is the incentive to create change? In urban environments, in which heritage is often employed as a marketing strategy, this is even more problematic—for in representing the city as "historic artefact", the city becomes a mere object, a spectacle, at the expense of the social fabric which it relies upon to sustain itself (Kafka 2018).

While nostalgia can therefore risk political paralysis, this is by no means always or automatically the case. Scholars have begun to push back against this gloomy characterization, redefining nostalgia as "productive" (Blunt 2003) and "progressive" (Smith and Campbell 2017). It has been advocated as a coping strategy in times of crisis, providing a sense of continuity and stability (Ritivoi 2002, cited in May 2017). Nostalgia does not have to be, as has been previously argued, a stagnating, conservative force but can provide the potential for change. As May (2017, p. 404) puts it, it can be seen as "a creative way in which people engage with changes brought on by the passage of time".

This acknowledgment of nostalgia as a productive force in the present is certainly a welcome development and helpful for understanding the ways in which a concern with the past does not automatically translate into a "melancholy, maudlin sentimentality and futile longing" (Smith and Campbell 2017, p. 612). Yet it is still fundamentally related

to an object of loss *in the past*. The object of anticipatory nostalgia, however, has not yet been lost; it is, rather, an imaginary state of *future* loss, lending it more of a utopian than melancholic quality. Grammatically speaking, this would be the form of the Future Anterior, or "what will-have-been", a way of imagining the act of retrospection from an imagined future standpoint. This notion of memory as a form of future projection is gaining traction in cultural studies (Barros 2021) and parallels recent scholarship that sees heritage practices as fundamentally future-oriented (Harrison et al. 2020). Photographs then, contra Barthes, do not only gesture to absence and loss, or even the past; they are simultaneously traces of physical reality as well as interpretive documents that speculate and postulate. Photography, through its creative powers of interpretation, synthesizes past memories and future desires to construct what Barros calls a "memory of the future" (2021, p. 155). To use David Bate's pithy phrase, the photograph becomes not a statement of the "what has been" but of "what will be" instead (Bate 2010).

What is most salient here is that home and feeling at home are not only spatial but *temporal* phenomena, and photography is an excellent medium through which to explore this. In the following sections, I want to unpack the ways in which home—and un-homing—is experienced as not only a spatial, but a temporal phenomenon, and suggest that the use of black and white in this context represents both a sense of loss, or a temporal rupture with the present and, paradoxically, a sense of belonging.

## 5. Out of Time, out of Place

Crucial to an understanding of nostalgia in all its forms, then, is a concept of temporality—a complex, ambiguous one at that. According to Batcho and Shikh (2016), anticipatory nostalgia is in some ways a double-edged sword, for while on the one hand it enables individuals to "reappraise" the present and/or future and can thereby provide a greater sense of meaning, it also runs the risk of "distancing" the self from the present, resulting in feelings of sadness caused by the foreshadowing of loss.

It is this idea of temporal distance, or disruption, which is especially pertinent to discussions of photography, memory, and home. If anticipatory nostalgia can be seen as an expression of feeling "out of time" as well as "out of place", then what are the implications for ideas of home and belonging? How might photography, especially in black and white, be a useful tool for mediating the tensions between temporal loss and distance on the one hand, and a reappraisal of the present and future on the other? How might this reappraisal address notions of cultural memory and identity? What I wish to draw out here is the fact that a sense of home, or loss of home, is *temporal* as well as spatial in nature; feeling at home is a feeling of being "in time" as well as in place.

The significance of temporality has been explored by gentrification scholars, notably Kern (2016), who argues that gentrification is characterized by a series of slow, incremental changes to the landscape which gradually lead to a sense of exclusion. Rather than focus on spectacular acts of displacement, such as state-perpetrated demolitions or large-scale urban renewal, she argues that it is these subtle visual changes which merit further attention. Photography, in its ability to freeze the current moment, is thus an ideal medium with which to document these details, visual clues which might otherwise go undetected. Again, because monochrome *already* exceeds our expectations of "realism"—we do not see in black and white—we pay closer attention to what is pictured (Freeman 2017).

Baldwin and Keefer (2020) also make a plea for the central role of temporality. They argue that while the spatial dimensions of belonging have been extensively researched, the temporal aspects of feeling at home have been severely neglected. To belong to a particular place, whether domestic dwelling, neighborhood, city or country, a sense of what they call "temporal rootedness", or being at home in the present, is essential. The authors argue that temporal rootedness matters because "People thrive in part when they feel at home both *here* ... and *now*" (2019, p. 3078).

## 6. Archival Objects: Cultural Commemoration in Black and White

If this is the case, then how are we to read images of gentrification that explicitly utilize an aesthetic of *pastness*? I argue that these images embody a fascinating temporal ambiguity. For on the one hand, the use of black and white signals the impossibility of return, a sense of melancholia, mourning, and regret—a temporal irreversibility and condition of loss. Home is relegated to the realm of "the past": another time. As argued in the previous section, this aesthetic of pastness, with its air of inevitability, carries with it the risk of rendering political action in the present inert (Kafka 2018). To what extent might this type of nostalgic visual practice be contributing to a kind of "museumification" of gentrifying areas? On the other hand, could it be *precisely* this pastness that lends these images their sense of significance in the present, bestowing on them the status of "archival objects"?

The archive as a concept is inextricably linked to processes of identity construction and feelings of belonging (Schwartz and Cook 2002). By enabling groups to locate their identity in a shared visual memory, the archive confers a stabilizing influence on collective identity in the present (Bate 2010). It is also crucially tied to the concept of power. Historically, only elites would have been able to create, maintain, and access archives, which they used to uphold claims of truth and legitimacy in order to secure their rule (Lowenthal 2007). Archives are no longer trusted as pure witnesses to historical fact, however, and this suspicion toward the impartiality of the archival record, along with the "memory boom" that has extended from academia to the public sphere and to the arts, has seen an explosion in the creation of independent archives and counter-memory projects (Flinn 2011; Carbone 2020). The archive may no longer pretend to any kind of historical objectivity, but as "a *symbol* for expressions of power, what is remembered or forgotten in society, and what is knowable and who has the power to make knowledge", its use as an aesthetic strategy is significant (Carbone 2020, p. 258).

I locate several examples of this archival aesthetic in the work of contemporary photographers; I have chosen to showcase three here. By performing a close reading of several images using a visual literacy methodology, I argue that these images communicate values that are archival and nostalgic, constituting an unofficial record of visual collective memory.

## 7. Reading Photographs: Key Considerations

According to Zinkham (2007), there are several key aspects to take into account when conducting a close reading of a photograph, including but not limited to the following: the purpose for which the image was created (the function of the photograph); the techniques used to produce the image, visual conventions, and (if known or relevant) audience reception. In short, a focus on the "context, contents and methods" used to produce the image (Zinkham 2007, p. 59). Zinkham suggests that by combining knowledge of the major visible elements of an image with knowledge of its purpose/function and visual techniques, a narrative interpretation of the photograph is made possible. Using the triad above, then, I will consider the work of three photographers who all communicate a sense of nostalgia for the present.

## 8. Lorenzo Grifantini

One photographer whose work is paradigmatic in this regard is Lorenzo Grifantini. As a longtime resident of the Golborne Road area in London, he has used his photographic practice as a way of archiving his local community before it is lost to gentrification. It is, as he describes it, "an act of love" (photographer's website, see references). The images below were all exhibited at the Portobello Photography Gallery in London in 2014.

With Zinkham's triad in mind, let us consider the image shown above (Figure 1). What do we see? What are the major visible elements in the picture? A group of women, dressed in the traditional hijab, walking together. Only one appears to have noticed the photographer; the others look down or in another direction. We see an iPhone, two bottles of water. The women take up most of the space, though we see a flyover in the background.

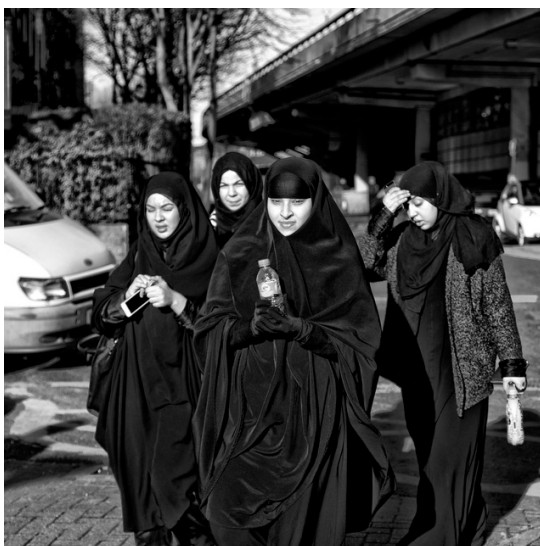

**Figure 1.** Muslim women gather to browse the market stalls. Image courtesy of Lorenzo Grifantini.

Consider a second image alongside the first (Figure 2). Men, seemingly colleagues or at least acquaintances, take up the left half of the image; one looks directly at the camera while another gazes into the middle distance. The camera's perspective makes us feel like a part of the group, lending a certain sense of intimacy to the shot. They all appear to be of African/Afro-Caribbean or Black British descent. In the background, two men shake hands; perhaps one is a customer. The group is standing before an uncovered street market stall packed with shoes and clothes on racks. Stalls stretch as far as the eye can see, beyond the frame, and as in the first image, we see the flyover more prominently featured this time.

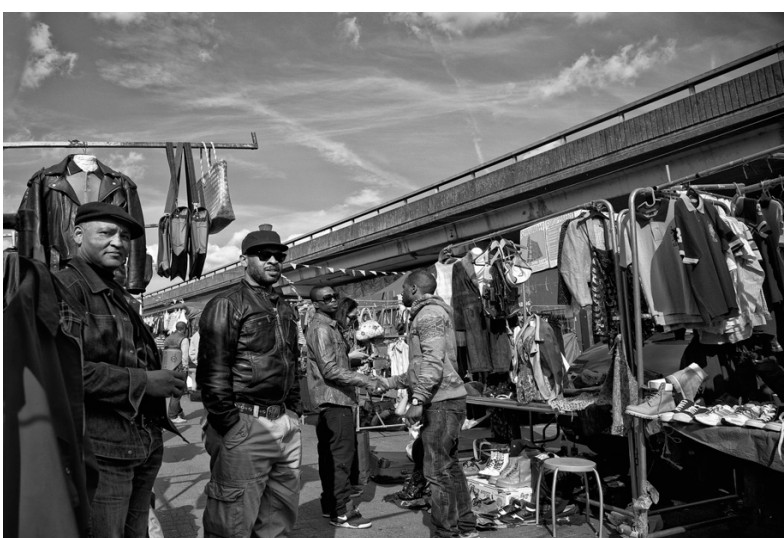

**Figure 2.** Market traders under the Westway flyover. Image courtesy of Lorenzo Griffantini.

As images always operate intertextually, informed by their positioning alongside one another and in relation to other images (Werner 2004), it is important to consider these two images as single components in a wider contextual narrative. First, how do they relate to one another? Arguably loosely, apart from the single repetitive element: the flyover. Second, how do they relate to the external context beyond the frame? What are these pictures "about"?

In order to dissect the contextual significance of these two images, we need to delve more deeply into the geographical—and simultaneously symbolic—site of their production:

Portobello Road. Portobello Road with its market, made famous by films such as *Notting Hill*, is arguably one of the world's most iconic market streets. Although controversies around gentrification are not new to the area, which has long been seen as one of the most desirable (and costly) areas of London, the market has been the subject of intense local protests because of proposals made in 2015 to extend the market and 'regenerate' the area. Various proposals (some of which have been dropped because of local opposition) have included space for an arts center, an indoor market hall, shops, cafes, more market stalls, and a "cultural space" "that can provide a broad spectrum of different art forms and create a sense of local pride in the heritage and creativity inherent to the area" (Bloomfield 2018, Evening Standard). The trust behind the development has argued that the market itself (particularly the section underneath the Westway flyover, pictured) contains pockets of space that are "tired and under-utilized" while local residents, who have petitioned the trust, argue that these spaces are "authentic" and are what give the market its "individuality" (McAteer 2015, Metro). What makes this area all the more significant is that it is only a short distance away from the Grenfell Tower, the public housing project that caught fire and killed 72 people in 2017. It is an area of London—like many—in which immense wealth sits in close proximity to extreme poverty.

Interestingly, the Westway Trust, which is responsible for delivering the regeneration, is explicitly utilizing "culture" as a means of regeneration, a tried-and-tested strategy in urban economic management (Miles 2013) as well as a longstanding theme in the gentrification literature (see Zukin 1995). They note in their 2019 (Westway Urban Design Strategy Report 2019) report that

> The area is rich in history, culture and traditions built up over many years . . . Culture can help to animate places and encourage people to visit, socialise and stay longer. It can also provide local people with the opportunity to identify themselves collectively and to continue to retain and enhance the distinct character and identity of the area.

This suggests that the local community currently lacks a means of collective identification and that the "distinct character and identity of the area" is somehow at risk if the proposals do not go ahead. They emphasize that the proposals can create "A place for everyone", "a place that could be made greener, healthier and more attractive for people to live, work, play and do business" (2019, p. 8). Yet research has shown time and again that these metrics of sustainability, health, and aesthetic beauty are exactly what end up making areas like these ultimately inaccessible to all those but the most well-off (Gould and Lewis 2017; Lindner and Sandoval 2021).

Grifantini's photos reject this narrative of a place that is "tired" and unattractive. Crucially, they show an area *already* thriving, alive with commercial and leisure activity, inhabited by the young and old alike and from a range of diverse backgrounds. The market stalls pictured above are predominantly owned and managed by a community of Caribbean migrants, who settled in Britain after the 1948 Commonwealth Act enabled all Commonwealth residents to live and work in the UK (Royal Borough of Kensington and Chelsea Website 2015). Images like these underline how part of the much-vaunted "distinct character and identity of the area" is found precisely in these mundane settings and activities, in which locals trade goods, children play, and people of diverse backgrounds engage in their everyday lives. In contrast to descriptions of the area as "tired", we see the flyover through Grifantini's eyes as an object around which commerce, play, and leisure take place. It is a site around which diverse communities—be they Muslim women or Afro-Caribbean traders—coalesce. Significantly, almost all of his photographs depict ethnic or religious minorities, some of which have often been pictured in more threatening or hostile ways in other visual media. Consider, for example, the visual politics of the hijab in tabloid media images. Grifantini, however, shows us a place that is already "for everyone", to paraphrase the Westway flyover report—subtly suggesting that precisely the opposite will happen if the market stalls are "regenerated". This is *already* a home in the sense of a

lived space and a space of belonging in which residents are connected to one another. What these images implore us to ask is, what is a home without its community?

This is where the visible content and the context of these two images—and the others with which they are displayed—come together with the visual technique of monochrome to create a powerful yet subtle argument. To recall Grainge's arguments on the nostalgic power of black and white, these images elevate their subjects; monochrome lends them a certain authority, acting as a signifier of "what should be considered historic business" here in the present, before these places are actually lost to the forces of gentrification (1999, p. 392). We, as spectators, feel both a temporally misplaced sense of nostalgia—missing the "death" of the neighborhood before it has occurred—and a recognition that through this act of visual commemoration, we or distant others might feel compelled to take action and prevent the loss from occurring. This action—performing an aesthetic revaluation of the area and its "tired spaces"—may only be symbolic, but it is one strategy in decoding the apparently neutral language deployed in reports such as the one cited above, in which references to the "character" and "identity" of the area are used without consideration of the fact that to those who live and work there, the area of Portobello Market and Golborne Road *already* possesses a unique and special identity. Indeed, it is precisely this that is felt to be under threat from the plans proposed in the report. Photography is crucial in contexts such as these, in providing a visual counter-narrative—and archive—of urban change.

### 9. Colby Deal

Another such project is Colby Deal's *Beautiful, Still (2022)*. Shot exclusively in black and white, the project documents the predominantly African-American Third Ward neighborhood in Houston, Texas. The photographs, which feature a mixture of people and landscape, aim to document the once-thriving community before it is lost to gentrification. Deal says of these changes that "Traditions are being erased. Characteristics are being erased. Buildings are being torn down. Homes are being torn down. Families are being forgotten about and broken apart. The artwork serves as that memory of the original people that were there, that are leaving now or being pushed away" (Magnum Photos Website n.d.).

These images are analog, rather than digital. The use of film, Deal says, forces the photographer to "really see" the environment he or she is working in, rather than the quick-fire ease of the digital image in which computers perform a large part of the photographer's work (ibid.). Not only does this demand a certain slowness and immersion, it also imparts a certain rough quality to the work, a nostalgic authenticity that rejects the perfection of digitally edited work. Additionally, there are no signs of modern technology or contemporary branded objects anywhere in the frame. These images, which preserve the memory of the Third Ward's traditions, buildings, and people, are later transposed on to large-scale outdoor installations all over the Third Ward and throughout the city using cheap, everyday materials—reflecting this visual archive back to the people who helped create it.

Many images are staged scenes of close-knit family life, with subjects seen listening to music, sitting around tables, and playing chess outside in the front garden or on the street—all powerful motifs of (Southern) African-American culture. Staging provokes an immediate response in the viewer; according to Deal, it "force[s] people to ask questions" such as whose image is being captured, why, where, etc. (Project Row Houses Website n.d., interview). Look at the image below, for example (Figure 3). We see two men, backs facing to us, deeply engaged in a game of chess (the second of Deal's images to show chess being played). Greenery takes up a large proportion of the space, lending a peaceful air. The men are seated on beautifully upholstered chairs, yet the grass is visibly overgrown; even more jarringly, the house behind them is boarded up with large wooden planks. A surface-level reading suggests a sense of leisure and conviviality, while a closer look reveals an unsettling sense of decay.

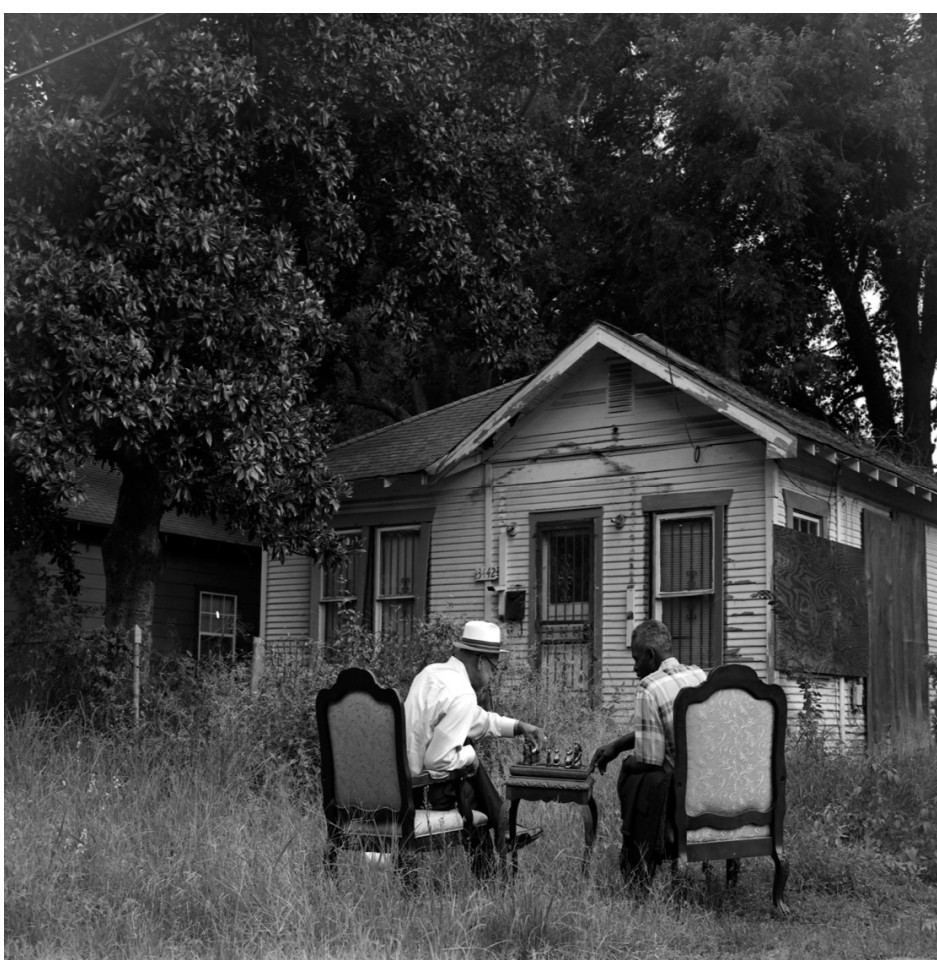

**Figure 3.** Playing chess. Image courtesy of Colby Deal. *Beautiful, Still*. Published by Mack Books.

In the following image, also staged, we see an African-American woman, posing in an elegant white lace dress for a portrait against a backdrop of what appears to be a boarded-up house (Figure 4). She is centrally positioned, looking down slightly into the camera's lens. The background is softly blurred, but it is just about possible to make out plywood planks and damaged fencing.

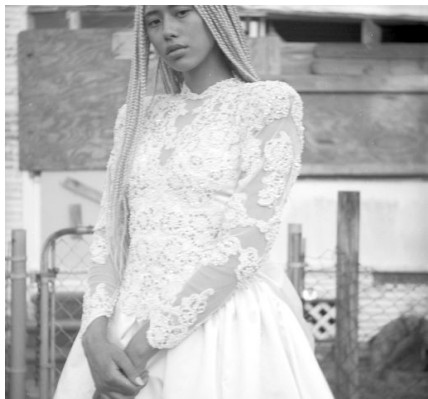

**Figure 4.** Woman in white. Image courtesy of Colby Deal. *Beautiful, Still*, published by Mack Books.

Decay and deprivation feature in other images, too. Below (Figure 5) we see two upturned shopping carts overflowing with trash. The sidewalk is cracked; there is other litter strewn across the ground. A sign is pictured for the Multi-Service Center & Health Center.

In all three images, Deal uses the aesthetics of decline, yet in ways that do not conform to any straightforward narrative. In images one and two, for example, he utilizes decay to challenge assumptions about his subjects. While images of ruined houses are everywhere to be found in contemporary visual culture, Deal uses the technique of staging to insert aesthetically pleasing and nostalgic elements into the frame. In the first two images, ornate furniture and clothing suggest a sense of tradition, formality, and pride in the subjects, despite their surroundings. The second image in particular disrupts conventions of portraiture (a genre historically associated with elite self-fashioning) to make a claim about Third Ward residents' identity in a context of decline (for more on portraiture see Johnson 2005). The third image seems to be more consistent with common conventions of ruin photography, with its lack of human subjects, and rather than refuting the narrative of decline, seems simply to acknowledge a sad contrast between grand social ideals and actual physical reality. This is evident in the prominent focus on the sign for the health center, in which ideals of health and vitality collide with the visible decay in front of us. Read alongside one another, a narrative of defiance emerges through these scenes—a beauty and a perseverance in the face of difficult circumstances, encapsulated in the title of the series: *Still, beautiful*. Here, then, is where context, recalling Zinkham and Werner, provides the key to understanding these images. The Third Ward, like many African-American neighborhoods, has long been the target of stigma, and Deal's images provide a means of constructing an alternative spatial identity. Crucially, Deal's photos were publicly exhibited all over the neighborhood in large-scale mural-type installations. In a context of deprivation, they therefore have the significant potential to reconstruct narratives of aesthetic value. As the artist puts it, "Even though this neighborhood and community are viewed as ugly or something unpleasant, there's still a bunch of beauty here to be seen" (ibid.). There is a melancholic as well as nostalgic aspect to many of the images, a beauty in the sadness.

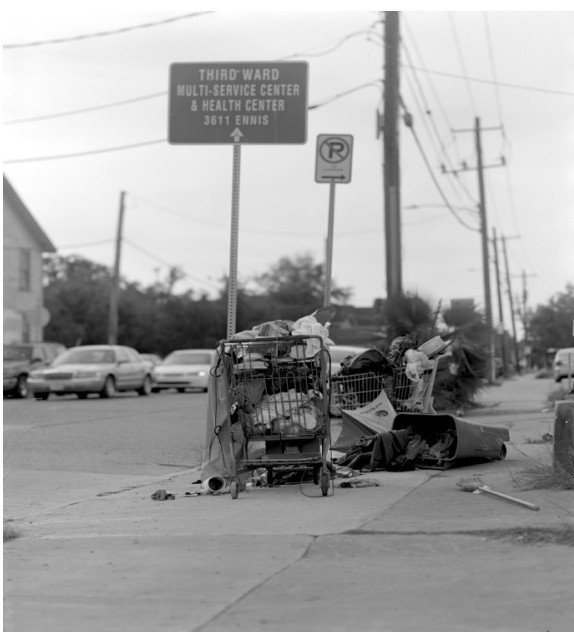

**Figure 5.** Overflowing trash. Image courtesy of Colby Deal. *Beautiful, Still*. Published by Mack Books.

As spectators, we occupy an uneasy stance, unsettled yet comfortably privileged in viewing the aesthetics of decline from the outside. This speaks to the ambiguity of nostalgia in these images, in which multiple and in some ways competing impulses coalesce. There is mourning for a home that is crumbling and physically vanishing, a backward-looking, melancholic attachment to the vanishing past, and a realization that this documentation can help to re-present the "ugliness" and stigma for a reimagined future. The representation of places and the people who live in them as existing in a state of erasure is an inherently

risky one, and any contextual analysis must consider the politics of this representation (Pusca 2014). As stated earlier, the aesthetics of nostalgia, as with all heritage-making techniques, can be viewed as a statement of inertia, a visual shorthand for places and people relegated to the dusty annals of history. It can also be seen as essentializing or reifying spatial identities. Questions of who creates these "archives", then, are particularly salient, and even more so when the creators in question come from outside the communities they are documenting. These concerns must be borne in mind. In terms of political inertia, however, and the question of whether such images can be of any relevance to the present and future as well as the past, the key point here is that nostalgia can be used not just to document—but to reframe and revalue. This reframing, a form of reimagining, makes sense if we are looking not just to remember "the past", but to impart *new meaning* to both the present and the future. We, as spectators, feel a nostalgia for the "will have been", a present that is gone by the time we look at these images, and similarly, a nostalgia for the future—an anticipatory nostalgia, an emotional experience of loss projected forward.

## 10. Jules Renault

A final example of this cultural revaluation through monochrome is Jules Renault's *Suspended in Time (2021)*. Unlike Colby Deal or Lorenzo Grifantini, Renault is not a resident of the area but works as an artistic collaborator with the musicians featured in his photographs—a local rap collective called the Bankai Fam. At their invitation, he spent years getting to know the area of New York known as Crime Heights and documenting the changes he witnessed.

As with Grifantini and Deal's images, I will apply a "content/context/method" approach here. In the first image below, we see a group of men standing in the doorway of a building. They appear to be engaged in some activity or otherwise distracted. One crosses an arm—perhaps defensively. Another is on the phone. The man in the background could be gazing at the camera, but he is blurred, so this is conjecture. The man on the left appears to be fiddling with some kind of fixture outside. The image appears to be spontaneous, un-staged; the subjects do not seem to take much notice of the photographer. In the second image below, we see a single subject—a man wearing a hat, necklace, and watch, sitting outside his home. He is blurred; he appears to be posing for the shot, but his features are difficult to make out. The third image is a series of images—a strip of negatives showing dogs playing, the dogs with two men who are presumably their owners, and finally two close-up shots of one of the dogs through a chain-link fence. If this is a superficial reading of the three images, what might a contextual reading reveal? As with the two cases above, the site of production—in this case, Crown Heights, Brooklyn—is key to understanding the purpose and function of these photographs.

Crown Heights, Brooklyn, was, according a 1985 *New York Times* article, "regarded for years as a dangerous ghetto", yet in a pattern followed by many other American cities, it has now become "the poster child for gentrification" (Hoffawer 2022). With rising rents, predatory landlords who allow buildings to fall into disrepair, and an influx of wealthy new arrivals, existing social housing residents (like those pictured in Renault's photographs, Figures 6–8) are sometimes evicted, find themselves having to live in untenable conditions, or are simply forced to move on (Chronopoulos 2020). It is hard to quantify just how many residents have been displaced; data show that some immigrant communities who moved to Crown Heights from the West Indies and Trinidad have gone back, while others have moved to other U.S. cities such as Boston, Philadelphia, or Atlanta (Yee 2015). Again, displacement can occur without any physical movement at all. This kind of displacement is symbolic and social as well as economic, leading to an uncanny sense of rupture or "un-homing", as it has been described (Elliot-Cooper et al. 2019). In the case of Crown Heights, this un-homing is drastically changing the character of a place that gave rise to one of the most significant periods of popular music culture: the birth of hip-hop.

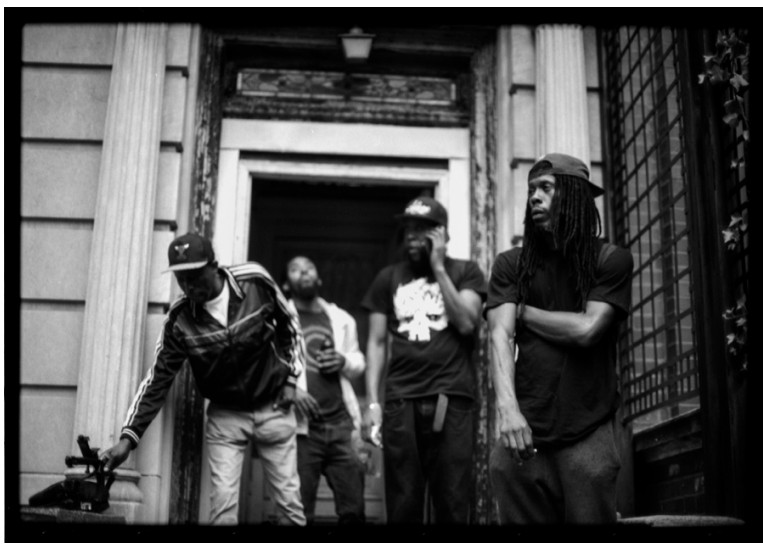

**Figure 6.** Gathering outside. Image courtesy of Jules Renault.

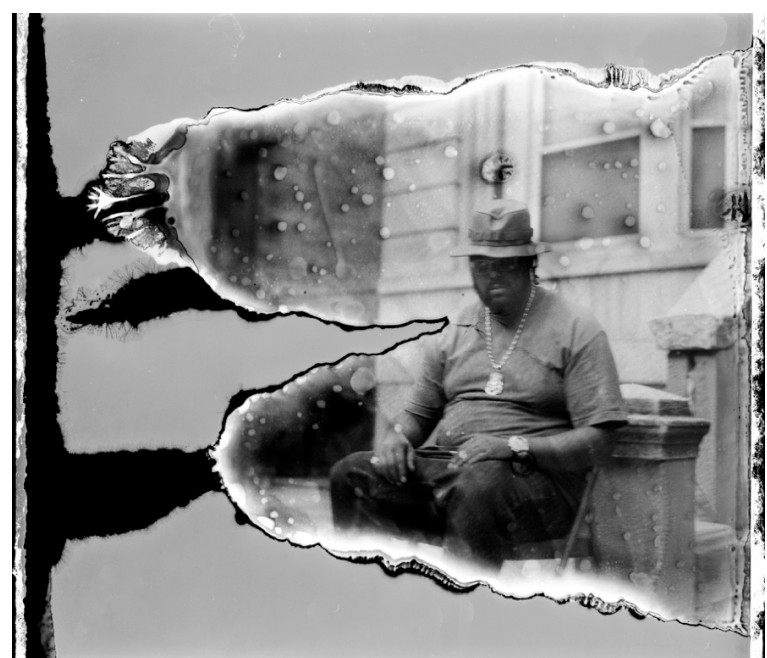

**Figure 7.** Man outside his residence, chemical drip. Image courtesy of Jules Renault.

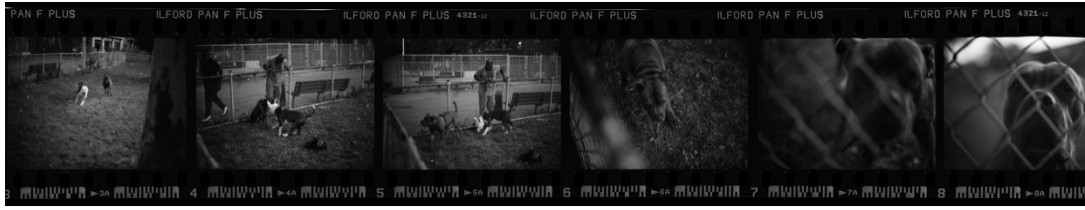

**Figure 8.** Dogs playing, negative. Image courtesy of Jules Renault.

Where Brooklyn could once lay claim to Jay-Z, Notorious B.I.G., Mos Def, Lil Kim, and many more major hip-hop names as former residents, "the word 'Brooklyn' now evokes artisanal cheese rather than rap artists", while streets that were named in rap songs as areas of crime and deprivation are "where cupcakes now reign" (Adler 2013). This influx

of wealthier residents, together with vehement opposition to new public housing developments (Fishbein 2016), means that the conditions that gave birth to a major movement in music history are likely to be eroded. The home of hip-hop is no longer recognizable and, according to Renault, is "bound to disappear" (photographer's own website).

Renault's images provide a stark contrast to this picture of Brooklyn. Often gritty and dark, they mostly depict the members of the Bankai Fam—the music collective who invited him into their community to capture these images. In other images in the collection, not pictured here but shown on his website and exhibited publicly in Paris, these men are shown sitting together, gathering on the steps of their public housing project, the Albany Houses. There are images of whiskey bottles, men posing on motorbikes. In this collection, there is no trace of cupcakes, no artisanal cheese shops to be seen. This kind of scene—in which young men socialize on street steps, drinking, watching their dogs play—is endangered, and this threat of erasure is directly linked to the loss of public housing projects in which these pictures were taken. These images of Crown Heights and its residents can be read in multiple ways: as a visual form of veneration, in which the spaces of hip-hop culture are documented and celebrated as significant, or, perhaps, less generously, as a form of aestheticization by an outsider, in which symbols of deprivation (bottles of alcohol crowded on tables, public housing infrastructure shown in dark, almost black tones, aggressive dogs jumping up at the viewer from a chain-link fence) are dramatized for aesthetic pleasure. The fact that Renault continues to collaborate with his subjects (shooting music videos, for example) suggests that they, at least, do not feel exploited in this way.

What is clear is that as with both Deal and Grifantini, the physical and social context of loss gives rise to an aesthetic deeply rooted in pastness—in the aesthetics of black and white. Again, all three cases show the use of monochrome as a form of cultural signification, of elevation to the status of memorial artefacts, of subjects who merit heritage status. Yet Renault's work utilizes a sense of nostalgia to an even more extensive degree than either Deal or Grifantini, in that he works exclusively with analogue processing techniques and film that is out of date or expired. His work, which is noticeably darker in color tone than in the first two cases, is full of chemical imperfections or "drips"; the loss of Brooklyn as a dynamic scene of artistic creativity and the displacement of the subjects he pictures is mirrored by the material transience of the images themselves. The use of negatives (shown in the third image above) as a form of visual technique and presentation is yet another instance in which the nostalgia of subject and form mirror one another. The material techniques of production thus work in tandem with the monochrome aesthetic to produce a sense of temporal loss and, as a consequence, of cultural value.

The prevalence of nostalgia in the contemporary cultural moment cannot be overstated, particularly in connection to the topic of gentrification. These bodies of work, which have all been produced in recent years, are also being circulated alongside a parallel wave of monochrome photography. These collections are "genuine" archival prints of "pre-gentrified" locations such as San Francisco, New York, several London villages—Dalston, Hackney, and Brixton—alongside many more examples too numerous to list. I do not think it is a coincidence that we are witnessing an aesthetic response to gentrification that is deeply rooted in nostalgia, either for the past—as in the examples described here—or for the present, as in the images shown above. Nostalgia offers us a way to mourn and an opportunity to act, and this action stems from seeing the world anew.

## 11. Belonging from Afar

In visually framing gentrifying places as "historic" sites, these images actively reconfigure the ways in which these places—and the residents who live there—are perceived. Gentrifying communities, often neglected for years and home to marginalized groups before being eyed up as profitable spaces by developers, are afforded a different narrative, an opportunity for visibility, a sense of agency in how they are portrayed, and a sense that they too are worthy of documentation and, even if only through the image itself, preservation. With their "archival aura" (Grainge 1999) these images help to construct

narratives of value, significance, and attachment to gentrifying places that is often lacking. Gentrification, which often relies on either a literal or metaphorical erasure—sometimes both—is antithetical to this sort of memory, attachment, and belonging.

Images such as those shown above demonstrate how the aesthetics of anticipatory nostalgia, underpinned by the archival connotations of monochrome, "enlivens" or "looks sideways" at the present by offering a glimpse of an imagined "future past" where gentrified places are given the cultural recognition they are currently denied. This narration of belonging "to another time", then, may instead afford another, more temporally ambiguous form of belonging that May describes as "belonging from afar". In this perspective, nostalgia becomes a form of "belonging in . . . if not to the present" (May 2017, p. 409), a phrasing that elegantly captures the ways in which multiple temporalities collide in the present to produce a stable sense of identity. Belonging, in this perspective, need not be rooted in the present; if time is a social construct, used creatively by individuals in times of crisis (Pulk 2022), then perhaps the aesthetics of pastness here in the present does confer a sense of belonging—just to a different time, narrated in the here and now. As Lewis and May argue, a sense of belonging " . . . can incorporate *a mixture of temporal horizons* as memories of the past and hopes for the future are used to enliven the present" (2019, p. 37, italics mine). The act of retrospection, or what Grainge (1999, p. 389) describes as "looking forward . . . in order to look back" can also, in addition to signaling a sense of loss, be a form of looking sideways at the present, and affirming its uncertainty over its inevitability. By looking ahead, it is the present that becomes "strange" (Rohrback 2015). It suggests, in other words, an openness or potentiality, a form of speculation and even hope as to future outcomes (Smith and Campbell 2017).

## 12. Conclusions: Reclaiming "the Aesthetics of Gentrification"

By challenging notions of the archive through the use of black and white photography, these types of images provide an important corrective to what Lindner and Sandoval call "the aesthetics of gentrification". Relying on a form of representation that promotes "seductive spaces", or spaces of "exclusive living and consumption", the aesthetics of gentrification often work to exclude and disempower marginalized groups (Lindner and Sandoval 2021, p. 14). This exclusion, I would add, is reinforced by a sense of forgetting and invisibility, as areas are redeveloped or newly built without any visible sense of shared memory or belonging over time. Home is marketed as little more than an economic asset, rather than a place to which individuals have developed close ties and to which they feel a strong sense of attachment.

Through making visible these ties and by framing gentrifying areas as *homes* in which certain communities have forged historical bonds of attachment, the aesthetics of black and white photography under discussion here provide a crucial counter example to the aesthetics of gentrification. These images reject such an aesthetics, positioning notions of home and memory as integral to representations of gentrification. Instead of a historical amnesia, this type of photographic practice and its archival aesthetics have been described as an important tool of preservation in these areas, granting the displaced a measure of agency, visibility, even belonging.

In addition, the use of black and white, which I have analyzed through the concept of anticipatory nostalgia, adds both a temporal and affective dimension to understandings of displacement in gentrifying areas. It underlines displacement as a condition of loss, even mourning, for places to which inhabitants may no longer feel a sense of belonging. Yet by appropriating the aesthetics of the archive, this type of photography also paradoxically communicates a sense of "belonging from afar". Finally, while a concern with memory has often been understood as a form of sentimental mourning for a lost past, this paper has emphasized the importance of an orientation toward the future. This perspective sees black and white photography as an expression of hope and a form of care for what Arboleda and Lorimer (2020, p. 30) poignantly describes as "places in the twilight hour". If, as Lindner and Sandoval (2021) argue, aesthetics is a key battleground on which struggles over place

and identity are played out in gentrifying areas, then I argue that is essential to attend to aesthetic strategies that articulate *alternative* visions of gentrification. For gentrification is not only about how places are made; it is also about how they are un-made or made unhomely through the displacement of people who once claimed those places as home. In refusing to forget, to inflict another form of loss through amnesia, these archival documents play a small but significant part in putting home at the center of the gentrification debate.

**Funding:** This research received no external funding.

**Conflicts of Interest:** The author declares no conflict of interest.

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
