# Peer review of "Missing the Present: Nostalgia and the Archival Impulse in Gentrification Photography"

_arts, 2023_

Round 1

Reviewer 1 Report

A thoroughly enjoyable read which covers a growing area of research surrounding the role of images, representation and aesthetics within how gentrification is visualised and contested.

I think the argument that monochrome/black and white provides a counter narrative to sites undergoing gentrification (and the work of Lindner & Sandoval 2021) is a valuable one but I think you could be more critical of how black and white images in this context are constructed and perhaps consumed. I suggest exploring more deeply the politics associated with black and white/ monochrome as a representational choice and how it intersects with gentrification. In particular I think there is a danger in assuming that all black and white photography associated with gentrification offers a chance for reflection and collective memory –  consider;

-          How does the use of black and white add or disrupt the sense of time/temporalitiy that is captured through the nostalgic aesthetic which is also reflected in the experience of unhoming. This links to concepts of slow violence, mourning and trauma.  (See Kern, L. (2016). Rhythms of gentrification: eventfulness and slow violence in a happening neighbourhood. Cultural Geographies, 23(3), 441–457. https://doi.org/10.1177/1474474015591489

-          I think your current argument could critique the role of black and white images as archival objects in greater depth –  the archive itself being a highly politicised space/experience. In this regard – by choosing black and white and curating an archival aura around images of gentrification how does this process change the meaning of what is being visualised? From your discussion and the distance and authority that you highlight is constructed through the archival aesthetic I would argue that this has a tendency to historicise the present moment which can make future loss feel inevitable. The transformation of space/people/communities as archival objects might then render the present political moment inert and reduce the space for people to challenge gentrification, as has sometimes been the case with other practices associated with Heritage Washing or Museumification where attempts to record or preserve the present become part of a wider gentrification process (see https://failedarchitecture.com/london-becomes-museum/

-          What are the politics associated with monochrome images that transform meaning around notions of the archive or historical document. What is lost or changed in this transformation? Similarly what are the power dynamics associated with who is able to construct this archival aura an the claims to knowledge they are making - are communities able to curate these archival images or are they purely dictated by the individuals that took them? Who is able to access them? What is excluded etc I would look at the wide body of literature around critiquing the archive and the feminist archive and explore how this might impact your argument.

-          It could be worth discussing the role of ruin lust or the aestheticization of poverty here (particularly for sites soon to become ruins or in a state of ruin awaiting gentrification), and perhaps unpack the role of black and white photography in structuring this process (See Fluck, W. (2010). Poor like Us: Poverty and Recognition in American Photography. Amerikastudien / American Studies, 55(1), 63–93. http://www.jstor.org/stable/41158482 ; Katie Beswick (2015) Ruin Lust and the Council Estate, Performance Research, 20:3, 29-38, DOI: 10.1080/13528165.2015.1049034 )

-          I felt the section Anticipatory nostalgia and the aesthetics of hope on p13 seemed a little out of place and perhaps repetitive – some of this could be split between your introduction where you introduce the concept and the conclusion where you can highlight its hopefully qualities. 

Another article that the author may find useful:

Candice P. Boyd & Andrew Gorman-Murray (2023) Nostalgia in black and white: photography and the geographies of memory, Australian Geographer, 54:1, 79-87, DOI: 10.1080/00049182.2022.2069330

Author Response

Many thanks for these much-needed insights. 

-On the topic of temporality: I have briefly outlined Kern's argument on slow violence, and specifically linked this issue to both photography in general and black and white photography in particular (line 254).

-On the risks of rendering the present inert: this was a valuable contribution, and one I have expanded on two or three times in the text as a result (see lines 198 and 559). 

-On the politics of the archive, I have included a paragraph on lines 555-565 on the importance of considering who is able to create an archive and how this undertaking is especially risky when conducted by outsiders. This point also returns in an expanded section on ruin aesthetics. Other points addressed include the present suspicion of archives and the adoption of an archival aesthetic by many contemporary artists as a way of probing issues of memory, knowledge and power. 

-I have added in some brief context on the controversies of ruin aesthetics and the wider politics of representing blighted communities (see paragraph around line 550).

-I have removed the section on Anticipatory Nostalgia at the very end to avoid unnecessary repetition. 

Author Response

Many thanks for these comments.

-In response to the primary concern stated in your review: I have significantly expanded my analysis of the selected images using a visual literacy methodology (content/context/visual techniques). I have also included some comments on the importance of intertextual analysis. This has enabled me to connect specific images to both the context of gentrification and to other images in the paper, leading to a much deeper- and more insightful- analysis. 

-On the second main concern: I have substantiated my claims about 'genre' and 'trend' in the introduction. 

-On the contradictions of photography as capturing 'moments' versus gentrification as a long process. This resonated with comments from reviewer 1 on the significance of temporality, and what photography can do in responding to shifts in temporality in the context of gentrification. I incorporated reviewer 1's recommendations on describing gentrification as a process of 'slow violence', and connected this to photography's inherent ability to 'freeze' these moments of change. I then linked this to expanded observations on monochrome's specific ability to make us 'step back' and scrutinise images (a revision to version 1). In other words, all photography is valuable as an instrument of documenting small subtle shifts in the landscape, and providing an archive of these changes; but monochrome does so with maximum impact. 

-On the use of words such as 'trend', 'genre' etc; I have included definitions of genre as texts which share certain communicative purposes, something which I found to be very useful and convincing in the context of monochrome. Please see introduction.

-On the need to provide a deeper analysis of photographic case studies. This section has been expanded significantly, and now includes components of both surface-level visible content and a deeper contextual analysis. I have been careful to elaborate on your excellent point on the multiple readings and slippery interpretations of photographs, something which helped me to conduct a more critical reading of these particular images. I have also as advised added in information concerning the dates of exhibition. 

-I have included references to Zukin and Miles on the use of culture as a regeneration strategy and removed the line you quote around line 58. 

-On the topic of how and if photographs can function as a 'call to action'. In light of reviewer 1's comments, I have been able to address this point more carefully. Specifically, the fact that black and white puts its subjects in the past risks a sort of inertia or political paralysis, and thus undermines the drive to take action in the present. I've also linked this to critiques of the heritage industry and its tendencies towards 'museumification' more generally. In short, I have been a little more cautious with this claim, so many thanks for these comments!

-Organizational improvements. Some paragraphs have been entirely removed and some combined to avoid repetition. Hopefully this has led to greater conceptual clarity.